# Analysis of the Development of Gender Stereotypes and Sexist Attitudes Within a Group of Italian High School Students and Teachers: A Grounded Theory Investigation

**DOI:** 10.3390/bs15020230

**Published:** 2025-02-18

**Authors:** Francesco Sulla, Barbara Agueli, Andreana Lavanga, Maria Grazia Mada Logrieco, Stefania Fantinelli, Ciro Esposito

**Affiliations:** 1Department of Humanities, University of Foggia, 71122 Foggia, Italy; francesco.sulla@unifg.it (F.S.); andreana.lavanga@unifg.it (A.L.); maria.logrieco@unifg.it (M.G.M.L.); ciro.esposito@unifg.it (C.E.); 2Surgical, Medical and Dental Department of Morphological Sciences Related to Transplant, Oncology and Regenerative Medicine, University of Modena and Reggio Emilia, 42121 Reggio Emilia, Italy; 3Department of Theoretical and Applied Sciences, eCampus University, 22060 Novedrate, Italy; stefania.fantinelli@uniecampus.it

**Keywords:** gender stereotypes, sexism, adolescents, trainee teachers, focus groups, Grounded Theory Methodology

## Abstract

Gender stereotypes and sexist attitudes continue to persist in educational settings, with significant implications for students’ achievement and well-being. This study aimed to investigate the development of gender stereotypes and sexist attitudes among Italian high school students and trainee teachers. A series of focus groups were conducted with a group of Italian school students and trainee teachers to uncover the complex interplay of individual, interpersonal, institutional, and societal factors that contribute to the formation and perpetuation of these biases. Analysis was conducted using a grounded theory approach. The findings reveal a nuanced and multifaceted understanding of the issue, highlighting the critical role of teacher attitudes, peer influence, and broader cultural norms in shaping students’ perceptions and behaviors. This study offers insights for educators, policymakers, and researchers seeking to address gender inequities in education and promote more inclusive and equitable learning environments.

## 1. Introduction

Over the past 20 years, psychology has deepened its understanding of the development of gender stereotypes and sexist attitudes and their effects on people’s well-being (e.g., [5]; [9]; [46]). Indeed, social systems such as families, peers, schools, and media may play key roles in reinforcing sexist practices in young people ([9]; [64]). The roots of this can be traced back to early development. For example, we often label activities as “for boys” or “for girls” even when it comes to toys for very young children ([5]), and same-gender peer groups encourage gender-conforming behaviors ([49]).

In time, cultural norms, including hegemonic masculinity, become harsh towards boys and girls, promoting, for example, male dominance and emotional suppression in boys ([18]) and submissiveness in girls. These gender stereotypes, rooted in traditional norms, create imbalanced relationships between men and women, often to women’s detriment. They are tied to notions of masculinity focused on power and control, which may harm both men’s and women’s psychological and physical well-being ([40]). The power dynamics associated with these gender roles may, ultimately, lead to violence and discrimination against women ([22]; [10]; [17]).

Scholars have suggested interventions to promote gender equality and prevent violence. However, many of these efforts focus on individual traits, overlooking broader contextual factors that shape social relationships and well-being ([27], [28]; [39]). This narrow perspective can lead to a myopic view, seeing the perpetrator of violence against women as a mere “monster” and the only culprit. Yet, this approach fails to acknowledge the collective responsibility of the entire Society. We believe a more effective approach is one that investigates the intricate interplay of multiple influences on people’s attitudes and behaviors, as the ecological approach does. This approach highlights the central role that education plays in perpetuating gender stereotypes. Indeed, cultural norms and educational models from parents, teachers, and the media reinforce gender-typing, teaching children to associate certain behaviors and roles with specific genders ([47]; [76]; [52]; [79]). These stereotypes begin in childhood and are reinforced throughout adolescence ([73]; [82]).

For example, when boys are encouraged to play with construction toys, they develop stronger spatial skills, which contribute to advantages in fields like geometry. In contrast, girls excel in computational tasks, but their math skills are often undervalued by parents (e.g., [33]) and teachers ([69]), contributing to a gender gap in future women’s representation in STEM fields. This gap is compounded by lower expectations for girls’ abilities, further limiting their confidence and interest in STEM careers ([43]; [21]). These stereotypes shape academic and professional outcomes, with women often underrepresented in high-paying careers due to persistent gender biases and cultural norms. Therefore, it is crucial to investigate the role of schools and teachers in shaping students’ attitudes and the transmission of gender stereotypes and sexist beliefs within educational settings.

This study aimed to explore the development of gender stereotypes and sexist attitudes among high school students and teachers in Italy.

### 1.1. Manifestations of Sexism During Adolescence

Traditionally defined as a system of prejudiced and discriminatory beliefs, attitudes, and practices that perpetuate the idea of male superiority and female subordination ([11]), sexism is characterized by bias or discrimination based on gender. It is rooted in social and cultural norms that establish distinct and hierarchical roles for men and women. This concept includes not only explicit manifestations of inequality but also seemingly positive attitudes that reinforce gender stereotypes. According to the theory of ambivalent sexism ([35]), the coexistence of power differences between men and women and their mutual interdependence creates two sexist ideologies that may appear opposing but are complementary: hostile sexism (HS), based on the belief that it is right for men to have more power than women, coupled with the fear that women may usurp this power by competing for it; and benevolent sexism (BS), which views women as pure, moral, weak, and passive, deserving men’s protection—as long as they conform. Hostile and benevolent sexist behaviors become more common during adolescence. Benevolent sexism often emerges in dating, with boys taking the initiative and paying for dates ([60]). Hostile sexism, such as sexual objectification and harassment, tends to increase from early to late adolescence ([9]). While both boys and girls experience these behaviors, girls are disproportionately targeted, and gender-nonconforming youth are particularly vulnerable to harassment.

Support for hostile and benevolent sexist attitudes among adolescents has been documented worldwide ([48]), but studies on ambivalent sexism in adolescents are less common compared to adults ([48]; [62]), primarily focusing on cisgender and heterosexual adolescents.

Research has explored how ambivalent sexist beliefs relate to behaviors like accepting or perpetrating sexual harassment ([62]). These attitudes have also been linked to negative behaviors toward sexual- and gender-minoritized peers ([12]), traditional views on heterosexual relationships ([7]), unrealistic romantic beliefs ([32]), and risky sexual behaviors ([62]). Moreover, girls with benevolent sexist attitudes tend to have lower academic or career aspirations ([30]; [67]) and are more involved in household chores ([53]), contributing to the perpetuation of gender disparities.

Considering these concerning findings, there is a critical need for interventions that address gender stereotypes and sexist attitudes at multiple levels, from individual mindsets to institutional practices and broader cultural norms.

### 1.2. Factors Influencing Ambivalent Sexism in Adolescence

Recent studies have explored how socialization experiences shape adolescents’ adoption of hostile and benevolent sexist attitudes. Research shows that ambivalent sexism is often reinforced by family influences ([38]; [57]), peer group norms ([26]; [54]), school practices ([19]; [20]), and media representations ([8]; [72]).

For instance, parents’ endorsement of traditional gender roles and lack of exposure to counter-stereotypical models can promote sexist attitudes in their children. Moreover, school environments where teachers and peers hold gender-stereotypical beliefs can also contribute to the perpetuation of sexism among students.

### 1.3. Sexism in School Context

Also, schools play a pivotal role in perpetuating or challenging gender stereotypes and sexist attitudes. Teachers’ biases, classroom practices, and curricular materials can all contribute to the development of sexist beliefs in students. Research suggests that teachers’ gender-based expectations and differential treatment of students based on gender, already present in earlier grades, may have long-lasting effects on students’ self-perceptions, academic engagement and achievement, and career aspirations ([70]; [69]).

Similarly, the gendered nature of school subjects, the underrepresentation of women in textbooks and teaching materials, and the lack of female role models in STEM fields can reinforce the notion that certain domains are more suitable for one gender over another (e.g., [13]).

Furthermore, incorporating gender-based stereotypes in textbooks, teaching materials, and instructional approaches can reinforce limiting notions of masculinity and femininity ([4]; [58]). Over time, analyses of educational materials have suggested curriculum changes, while research on the hidden curriculum has highlighted differential treatment and gender-based expectations (e.g., [36]). Reports have also noted violence, heterosexism, and insufficient support from teachers (e.g., [25]).

Moreover, according to expectancy–value theory ([80]), individual differences in students’ motivational beliefs are influenced by their experiences within school contexts. [66] ([66]) suggest that teachers can create opportunities for students to engage in a variety of STEM and non-STEM activities, which in turn provide students with information about their competence and emotional memories of these activities. Over time, these experiences shape the development of students’ competence beliefs and task values, which then influence their engagement in educational activities and future academic and career aspirations. These gender stereotypes and biases contribute to men being underrepresented in art, humanities, and care-related careers, and women being underrepresented in high-paying careers, highlighting the crucial role of schools and teachers in shaping students’ attitudes and the transmission of gender stereotypes and sexist beliefs within educational settings (e.g., [78]). Indeed, in the long run, these practices can contribute to the development of ambivalent sexist attitudes among both students and teachers, perpetuating a cycle of gender-based discrimination and inequalities.

### 1.4. The Gender Perspective in Initial Teacher Training

Initial Teacher Training programs play a crucial role in shaping educators’ gender beliefs and equipping them with the knowledge and skills to promote gender equality in their classrooms. Teachers’ resistance to gender issues might be linked to the training they receive, which often lacks strong policies or strategies emphasizing gender awareness ([65]). In Europe, gender perspectives in teacher training are minimal and often rely on individual initiatives or student interest rather than institutional commitment ([29]; [37]).

Research on gender equality in teaching faces limitations, including small sample sizes and a focus on voluntary participants, which skews results ([31]). Few studies show the impact of thorough training on reducing sexist attitudes ([51]). Thus, integrating a gender perspective in Initial Teacher Training is essential to improve education and counter persistent discrimination ([34]; [83]).

A recent study on Initial Teacher Training reveals inadequate attention to gender issues despite alignment efforts with the European Higher Education Area (EHEA). This cross-sectional study of 1296 primary teacher trainees used Ambivalent Sexism theory-based questionnaires and found that only female students showed reduced sexist attitudes. The findings highlight the need for comprehensive reforms to include feminist content and proactive gender initiatives in teacher training. Real educational change remains unlikely without addressing these attitudes, as teachers play a key role in fostering equality ([19]).

The gaps in the literature highlighted the need for a more comprehensive and in-depth understanding of how gender stereotypes and sexist attitudes develop and manifest within the specific context of Italian high schools.

In addition to the interactions with their parents, the most frequent interactions within adolescents’ microsystems occur with their teachers and peers. Therefore, this research aimed to explore the primary social beliefs and representations that might contribute to the creation of gender stereotypes and sexist attitudes in adolescents and the figures responsible for their education.

## 2. Materials and Methods

### 2.1. Participants

This study included two participant groups: 24 high school students and 24 trainee teachers. The students were all 18 years old, while the teachers ranged in age from 37 to 61, with a mean age of 43.2 (SD = 3.52). A gender disproportion was evident in both groups, with 87% females among the students and 79% females among the trainee teachers.

### 2.2. Procedure

Students have been recruited through convenience sampling at the University of Foggia during university vocational guidance meetings aimed at students in the fourth and fifth years of high school in the province of Foggia. During the meetings, the attending students were invited to participate in group interviews using the focus group methodology ([1]; [50]).

Convenience sampling was also used for the trainee teachers. They were recruited from among those attending special educational needs teacher training courses at the University of Foggia. During some of these classes, trainee teachers with prior experience as traditional high school teachers were asked to participate in focus groups, which were led by senior researchers who were qualified psychologists and supported by PhD students acting as observers.

The entire procedure was approved by the Ethics Committee for Research in Psychology of the University of Foggia (n prot. 014/CEpsi; 2 May 2023).

### 2.3. Instrument

The focus groups were designed and implemented similarly for students and trainee teachers.

Before conducting the focus groups, the research team (i.e., the authors) extensively discussed the topic before developing a grid of stimuli to propose to the participants in order to achieve the research objective.

Specifically, the grid—the same for both groups—addressed five thematic areas connected to gender issues: diffusion and relevance of the issue in the school environment; knowledge and experiences of unequal treatment based on gender implemented by parents and/or teachers; unequal treatment based on gender in peer interactions; possible consequences of gender-based unequal treatment; possible strategies to counter gender-based unequal treatment.

The students were divided into three groups, each consisting of eight members. Similarly, three focus groups were formed for the teachers, with eight participants in each.

The focus groups were conducted in university classrooms with chairs arranged in a circular format, as [44] ([44]) recommended, to facilitate dialogue and interaction ([45]). Senior researcher psychologists led the focus groups, with a PhD student observer supporting each group. The observers, in agreement with the participants, recorded the group interactions using digital devices, transcribed the recordings, and prepared activity reports.

Following [42]’s ([42]) indications, each focus group lasted an average of about 70 min and was subsequently followed by a careful analysis of the data that emerged by the conductor and the observer.

### 2.4. Data Analysis

Transcripts from focus groups from both groups were collected and analyzed together using an approach based on the Grounded Theory Methodology adopting a constructivist lens (GTM; [15]; [56]). The research team adopted an iterative, reflexive process for collecting, analyzing, and interpreting qualitative material.

The GTM involves the execution of three phases of analysis: a first phase of open coding, in which researchers assign an explanatory code to each portion of text that has meaning (quotation), consistent with the research objective; a second phase of axial coding, in which the research group reviews the codes attributed to the portions of text and groups those considered to pertain to the same concept in categories of commonality; a third phase of theoretical coding, in which the categories produced are in turn grouped into macro categories, which can summarize the results identified from the data.

Finally, the final step of the GTM consists in the creation of a core category, that is, a conceptual category, which is developed through a process of creative abduction performed by the research team, which represents the central hub around which the connections between all the identified categories develop ([55]).

To ensure the reliability of the data analysis, the research team discussed the codes and categories attributed, involving both the main researchers and the PhD students who acted as observers.

## 3. Results

The analysis conducted through GTM led to the creation of 291 codes, which were subsequently collected in 11 explanatory categories. Once the categories were defined, they were grouped into broader categories, which offered an overall understanding of the information that was identified from the interviews. After careful work, we defined four specific macro categories: Past, present, and future of sexism; Sexism in microsystems; Forms of sexism; and Consequences of sexism (see Table 1).

### 3.1. Past, Present and Future of Sexism

The first macro category “Past, present, and future of sexism”, refers to the temporal dimension in which the participants positioned the phenomenon of sexism, recounting their perceptions of its historical origins, current manifestations and possible future developments.

It is divided into two main categories: “Evolution of perceptions and experiences regarding gender inequality” and “proposed solutions to solve gender inequality and sexism”. These categories represent both lived experiences and potential strategies to address sexism and gender issues embedded in social and cultural dynamics.

The category “Evolution of perceptions and experiences regarding gender inequality” includes codes that highlight the respondents’ level of awareness regarding sexism and gender discrimination; the category also examines how people perceive and experience gender inequality over time, considering various factors such as personal experiences, the influence of socio-cultural contexts, and generational differences.

The most significant aspect of this category lies in the perception of sexist attitudes, which are frequently associated with older generations or backward socio-cultural contexts. This perception underscores the idea that societal and generational shifts are pivotal in shaping views on gender equality.

In fact, a teacher said: “*Young people are definitely more forward-thinking compared to adults. For example, I found myself discussing this with my 11-year-old niece, who agreed with the idea that women should be free to be themselves. My mother, on the other hand, was somewhat opposed because there was this topic about women, that to be a woman, you must necessarily be married, have children, and follow almost a standard scheme. And I noticed that my niece, despite being much younger, had a mindset that I found admirable—she was much more open-minded than an adult, who should ideally be a role model. Of course, nothing against my mother; she’s from another generation, so in a way, she can be justified.*” (P4, FG4, female teacher).

Older generations may exhibit behaviors or hold beliefs that reflect a time when gender roles were more rigidly defined, contributing to stereotypes and prejudices that younger generations may find outdated or inappropriate. Explanatory are the words of a student: “*The difference between our generation and that of our parents is precisely this: that we are open to question ourselves, to question our ideas, and perhaps even to change them in comparison with others; they may not […] Instead, we can change the rest of the generations that follow; however, now the generation of our parents…our grandparents…* [they] *cannot change their mentality, so in my opinion, this gap will remain for years to come.*” (P2, FG1, female student).

The second category, part of the overarching “Proposed solutions to solve gender inequality and sexism”, focuses on strategies suggested by participants to address and reduce manifestations of sexism and gender discrimination within society.

The associated codes reflect various approaches, from individual to collective, from education to social awareness. “Individual paths based on communication and information” and “not being influenced by others’ judgments” highlight a personal approach centered on self-determination and empowerment, ideological changes, and increased awareness.

“School interventions”, including “development of critical thinking” and “free expression of oneself”, emphasize the crucial role of education in promoting a more equitable and respectful culture.

These are the words of a teacher on the subject: “*My idea is to educate, in general, about freedom of expression, not only in reference to gender equality, but in every way, including being free to show who you really are.*” (P3, FG5, female teacher).

Codes such as “new education from the family on gender roles” and “youth should carry new values” suggest a generational shift in the transmission of values; while “For older generations, there is no hope” reflects a certain pessimism towards older generations.

The words of a female student are indicative of this theme: “*In my opinion, we need to replace values in which we, as a generation, no longer recognize ourselves. We should introduce a new type of mentality into families and schools. We need to work on the new generations. It is easier to change the minds of children than adults*” (P1, FG2, female student). This brings into play gender education, which is still underdeveloped and not incorporated into Italian school programs.

The participants’ proposals offer hints on diverse strategies that would promote a more equitable society while acknowledging structural and cultural challenges. It seems like they all agree that individual and collective transformation is essential for reducing gender inequalities.

### 3.2. Sexism in Microsystems

This second macro category, “Sexism in microsystems” focuses on the role of the various relational contexts, primarily the family and scholastic context, in the construction and transmission of sexist attitudes and beliefs.

Starting from the category “Gender issue in family”, we can note that this consists of a set of codes that refer to how gender roles and stereotypes are transmitted through family models, mainly based on the biological sex of children and parents.

In fact, the most frequent code for students is “disparity of treatment based on the gender of children”, while the most common code among teachers is “influence of the family on the mentality of children”.

Given their ages, the students reported their experiences and perceptions as children, while the teachers referred to their vision of their students’ parents, also influenced by their own experience as parents.

Significant, in this sense, are episodes reported by the students in which they felt they were treated in a certain way by their parents because of their gender. For example, females are given more restrictions on when they can go out, (as they say) to protect their safety. At the same time, males are granted more “freedom”, such as the possibility of staying out until late at night or being able to drive in the evening without necessarily being accompanied by other friends.

Explanatory in this sense are the words of a female student: “*As the younger sister with two older brothers, I’ve experienced a distinct difference in how my parents treated us. My brothers could do things like go out late at night, whereas I was not given the same freedom. I also felt more scrutinized, with my parents asking me many questions, while my brothers didn’t face the same level of scrutiny. […] Even simple tasks, like clearing the table, seemed to fall more on me, not necessarily because I’m a woman, but more as a family tradition or habit.*” (P1, FG1, female student).

The second code, as mentioned, refers instead to teachers’ perceptions of their students’ relationships with their parents and how these vary according to the student and parent’s sex.

As an explanatory example, we report what a teacher said: “*A boy published an article in the school newspaper in which he proposed to establish gender-neutral toilets, and this raised a lot of criticism from a boy, who has a homophobic father, who began to insult him. […] I reckon that family context shapes boys*.” (P4, FG6, female teacher).

Moving on to the second category included in this group, namely “Gender issues at school”, here are all the codes that explain how teachers deal (or do not deal) with gender differences at school. We find, in teachers, codes such as “teacher training on the topic of sexuality” or “difficulty in dealing with sexuality in school”. Nevertheless, codes that describe a difference in the attitudes and behaviors of teachers due to the gender of the students are also included in this category. In fact, among the students, codes such as “a different attitude of a teacher for male and female students” and “teacher nicer with girls” are identified; they are indicative of the different approaches that teachers can sometimes have towards male and female students, putting into practice more or less conscious gender stereotypes.

Here is an example from the words of a female student who used them to describe the attitude of a male teacher at her school: “*Let’s say that he is very shy towards the students. So, he has a kind attitude towards the girls: if something happens with the boys, he gets angry immediately, while with the girls, he says, ‘Come on girls, let’s avoid it!’ […] He has a calmer attitude with the girls*.” (P1, FG1, female student).

In conclusion, the last category of this group is “Gender issues at work”, which includes all the codes that refer to the representation of women in the workplace. These codes primarily address the experiences of trainee teachers, who are already familiar with the world of work. Among these, the code “women discriminated against in the workplace“ reflects both the participants’ personal experiences and incidents they have observed.

This is exemplified by the testimony of a teacher: “*In the business context, there have been episodes of discrimination against women who, at the time of hiring, were rejected precisely because they are women.*” (P6, FG5, female teacher).

However, this category also concerns students and their ideas of women’s position in the workplace. In fact, some codes refer to the horizontal segregation of women, such as “the caring role of women in the workplace” and “women work mainly in part-time jobs”. For example, a student said: “*Because of the education they receive from their family and society, it is as if girls are already led to look for a job where they work fewer hours, because then they will have to look after their children. In addition, these are usually caring jobs, such as nurse and teacher. It is as if they could not do anything else.*” (P2, FG3, female student).

Finally, this category also includes the consideration of work–family balance, with students drawing attention to the need for a family to have an equal distribution of household tasks. One of the most significant codes of this category is, in fact, “dividing household tasks based on the importance of the work (and not gender)”. This code recalls the fear of adolescent females of having to deprive themselves of one of the two aspects once they become adults, and of being forced to make a choice between family and work.

As an example, we report the words of two female students that are very explanatory: “*Personally, as I aspire to become a doctor, I do not foresee having children, as I am aware that it would deprive them of the time and attention they deserve*” (P4, FG3, female student) and also “*In the end, I believe one must carefully consider the life they envision for themselves. If I remain unmarried by the age of 30, I do not view that as a problem. My priority is to first focus on achieving independence, advancing my career, and establishing myself professionally before considering other aspects of life. I do not perceive this view as something negative*.” (P3, FG3, female student).

### 3.3. Forms of Sexism

The macro category “Forms of Sexism” is divided into three main categories: “benevolent sexism”, “hostile sexism”, and “internalized sexism”. Each represents a different facet of sexism and describes the different ways in which sexist attitudes and behaviors manifest themselves, both in terms of their explicit and implicit expression.

Based on the theoretical distinction between benevolent and hostile sexism, various codes belonging to this macro category have been classified.

The “benevolent sexism” category includes codes referring to behaviors and beliefs that often appear and are perceived as being kind and protective but implicitly reinforce the idea of women as subordinate and dependent. This form of sexism is insidious also because, most of the time, even women do not perceive it as sexism at all. In fact, especially among teachers in our sample, among the most frequent codes were “no perceived experience of discrimination” and “no observed experience of discrimination” said by women. In this regard, some of the phrases expressed by the female teacher participants in our sample were: “*…but in all my university years I studied ‘peacefully’, I didn’t notice any differences in treatment [compared to their male colleagues].*” (P1, FG5, female teacher); “*… I’ve never observed nor being subjected to sexist attitudes at work*”; “*the gender equality discourse… at least as I understood it… I have never had first-hand experience…*”; “*In my experience I do not see this disparity between them [referred to her students]*”. (P1, FG6, female teacher).

Furthermore, among the most emblematic codes in this category are: “men feel obligated to pay the bill”, “the notion of women as needing protection”, and “men and women are different but complementary”. These codes reflect attitudes based on a paternalistic view of gender relations, in which women are portrayed as fragile or incapable of acting independently. One male physical education teacher in our sample, for example, reported that “*it happened to me sometimes, to feel more protective of girls, to tell the truth […] while doing activities that are very physical, I tend to worry and say ‘be careful’ to boys ‘if not, you’ll hurt your [socialised as/female] classmate*.” (P3, FG4, male teacher).

The concept of protection is often disguised as “chivalry”, obscuring the fact that such practices presume an imbalance of power and competence between the sexes.

In contrast, “hostile sexism” manifests in overtly devaluing or discriminatory attitudes toward women. This discriminatory attitude also extends to the workplace, where women are often penalized when competing for stereotypically male roles. In our teacher sample, some of the participants reported being discriminated against in their previous job (e.g., architect, engineer)—and did not perceive being discriminated in school instead (e.g., “*this, however, only concerns the school context, because in the corporate context there have been incidents of discrimination against women who were rejected at the time of recruitment precisely because they are women*” (P6, FG5, female teacher).

However, some teachers also reported being discriminated against at school, even from their students. For example, one reported: “*one girl looked at me and said ‘it is not our fault if professor* [male name] *can keep the class quiet and you cannot’.*” (P2, FG5, female teacher).

In summary, hostile sexism not only perpetuates the view of women as subordinate but also fosters discriminatory and violent behaviors. Its social and emotional implications highlight the need to counteract these attitudes to promote greater gender equity openly. Codes such as “some sports (e.g., soccer, basketball) are more suitable for males”, “women can’t drive”, and “social media content overtly sexualizing the female body” reflect a societal narrative in which women are consistently placed in a position of inferiority. This type of sexism is particularly evident in the representation of women on social media, where the female body is often objectified and instrumentalized. The influence of stereotypes extends to language, as seen in the disparity of certain feminine terms acquiring pejorative connotations. Such dynamics not only damage women’s image but also perpetuate a culture of exclusion and oppression.

Finally, “internalized sexism” represents a less visible but equally significant dimension of the phenomenon, wherein women, influenced by cultural and social messages, unconsciously reproduce sexist attitudes toward themselves and other women. Among the most representative codes are “women posting photos in revealing clothing are judged (especially by other women)”, “women judge other women’s sexual behavior”, and “I won’t have children because I want to be a doctor”. These codes reveal how internalized sexism can manifest through social judgment and the limitation of personal choices, reflecting a deep internalization of sexist beliefs.

A particularly interesting aspect is the self-censorship some women apply to their aspirations and behaviors, such as perceiving motherhood as incompatible with a professional career. As reported by adult interviewees, this phenomenon often extends to their daughters and students. Sometimes unconsciously, they tend to guide adolescent girls toward options deemed as “more appropriate” for their gender, thereby narrowing their future opportunities.

The overall analysis of the “Forms of Sexism” macro category highlights the complexity of the phenomenon, which is rooted not only in explicit social attitudes but also in more subtle and pervasive dynamics. The three categories interact, creating a system that reinforces gender inequality through social norms, stereotypes, and internalized processes.

### 3.4. Consequences of Sexism

Finally, the fourth macro category, “Consequences of sexism”, includes a range of dynamics, experiences, and reflections related to the impact of sexism on the daily lives of women and men, highlighting how gender bias can limit individual freedom, influence choices, and damage self-esteem. These codes explore both the negative effects of sexism on women, often subject to discrimination and judgment, and those on men, who find themselves trapped in traditional stereotypes of strength and invulnerability.

The category “Consequences of sexism for women” describes the direct impact of sexism on women’s daily lives and emotional experiences. Codes such as “Sexist behaviors ‘hurt’ girls” and “Experiencing sexism lowers self-esteem” show how discrimination and microaggressions can negatively affect psychological well-being and self-perception. The psychological impact of sexism results as a dominant theme. These codes reveal how repeated exposure to gender discrimination may influence self-concept and individual aspirations. Here are the words of a student: “*I think that each of us, when we are judged, loses self-esteem. And to avoid it, maybe we tend to change the way we dress or a particular attitude, as we start thinking it can be inappropriate. And even if we are perhaps aware that we are not what others say, for fear of judgment, we are forced, let’s say, to change ourselves and this is very sad.*” (P4, FG1, female student).

The professional consequences of sexism also affect teachers and students in our sample, even though they are still distant from the labor market; many educators have reported direct experiences of gender discrimination in hiring processes and career progression. Through the code “Internalizing stereotypes as a result of discrimination”, it is highlighted that certain stereotypes persist in society to the point of being internalized even by women themselves.

To maintain control over women without constantly resorting to force, patriarchy must secure the majority’s consent, offering each woman an advantage denied to others. Some codes, such as “Walking alone at night is dangerous” and “Women often feel judged for how they dress or wear makeup” highlight women’s challenges in social contexts, particularly regarding personal safety and aesthetic judgment.

The perception of vulnerability in public spaces, especially at night, underscores a gender disparity in freedom of movement, often influenced by ingrained sexist stereotypes and behaviors. Similarly, aesthetic pressure translates into social control over women’s bodies and appearances, limiting individual autonomy through judgments that influence behaviors and personal choices.

Adaptive or defensive strategies result in response to such challenges, as the code “Indifference as a defensive reaction to catcalling” suggests.

This attitude represents a psychological mechanism for managing situations perceived as intrusive or potentially threatening, showing how many women are forced to develop emotional tools to cope with often hostile environments. Especially in these cases, both female teachers and students in our sample seemed to justify/accept forms of benevolent sexism linked to protection. For example, some of the female students reported: “*I like the gesture of the guys who make us stand on the inner side of the pavement (the one furthest from the road) to protect us… but also when at the bonfires they take care of the fire, and we take care of the food and crockery… it makes a bit of sense to me*”. (P5, FG1, female student).

The category “Consequences of sexism for men” explores the impact of gender stereotypes on the male population. Codes such as “Men are forced to fit into traditional stereotypes” and “A man must demonstrate his strength and worth” reflect the social pressure that compels men to conform to rigid models of masculinity. These stereotypes can generate emotional distress, as highlighted by the code “Some men suffer because they are not allowed to show vulnerability” indicating that men, as well, face limitations and pressures imposed by societal expectations. This aspect is clear in the words of one student: “*In our culture, a man cannot cry, a man cannot show emotions, a man must be masculine, he must show something, he must prove himself. This makes you feel inadequate and under pressure*.” (P4, FG3, female student). The social mandate requiring men always to appear “strong” and emotionless not only limits their ability to express what they feel freely but also confines them to a rigid behavioral model that denies their vulnerability. This constraint may negatively affect emotional management, leading to disproportionate reactions in stressful situations or intense emotions. The inability to recognize and address such feelings could result in aggressive or self-destructive behaviors, perpetuating stereotypes of toxic masculinity and creating difficulties in interpersonal relationships and mental health. While male teachers in our sample seemed more aware of the consequences of sexism for men—especially when talking about their students: “*we do notice that some male students suffer, for example, from bullying, but they open up more difficulty than female students*” (P4, FG4, male teacher)—younger generations seem less aware or it might be harder for them to show that they care. However, at the same time, they do not feel confident to cope with these kinds of issues raised by their students’ behaviors. The same teacher, at some point, said: “*I don’t want to open a Pandora vase, I’d prefer them to talk with the school psychologist*”. (P4, FG4, male teacher).

A male student in our sample, for example, complained about receiving less attention compared to his female mates “*it is well known that girls are more likely to receive attention than boys…for example on social media…we receive attention only if we adhere to a very specific physical stereotype (muscular, tall and well-built) […] a girl can present herself in any way, even without a sculpted abdomen, and still receive attention, whereas the boy necessarily needs physical characteristics to be noticed*”. (P6, FG1, male student). The same student at some point also said that he finds that there is also unequal treatment in employment and said “*it was difficult for me to find a job as they preferred girls for front-office roles within the catering industry*”. (P6, FG1, male student). The difference between male students and their teachers here might be because it is harder for younger boys to go beyond their personal experience and see the bigger picture.

Finally, the category “Female empowerment” includes codes emphasizing the value placed on women’s personal and professional autonomy. Among the most significant codes is “I don’t want to be a kept woman”, which highlights a strong desire for emancipation from traditional roles imposed on women; this illustrates how social norms and professional expectations intertwine to create barriers that discourage women from aspiring to leadership positions, thereby perpetuating occupational gender segregation. Moreover, it reflects young girls’ awareness of future expectations reserved for them as women.

Explanatory in this regard is what a female student said: “*There are still many aspects to change, from a social and cultural point of view I mean. There is still no equality today. But things are changing a little, and I personally do not intend to be like my grandmother or my mother. I want to be a different, emancipated woman.*” (P5, FG1, female student). Society, especially at the local level, predominantly portrays women as homemakers lacking personal income, which would grant them a certain degree of autonomy. However, this response from a female student underscores a desire and/or need for perspectives beyond those traditionally reserved for women. The code “Family upbringing fostering independence” underscores the significance of family-based education promoting female autonomy. Codes like “Work makes women independent” and “Influencers as symbols of female entrepreneurship” highlight the perception of work as a fundamental tool for building an independent identity and overcoming cultural and social limitations.

### 3.5. The Core Category: Ecology of Sexism

Conceptualizing the data led to identifying a core category, “Ecology of sexism”, as the central hub around which the connections between all the identifying categories develop.

The ecology of sexism is a concept that allows us to understand how sexism develops, manifests, and perpetuates itself through an ecological model. This systemic approach is derived from Prilleltensky’s ecological model (2007), used in psychology to analyze individual interactions and environments at various levels. In this case, the idea is to map how sexism results from interconnected individual, relational, community, and societal factors.

This involves representing sexism not only as an issue of discrimination but as a system that impacts individuals and relationships, proposing pathways for social and personal transformation with an intergenerational perspective—focusing on the intergenerational transmission of sexism and stereotypes.

## 4. Discussion

The findings of this study provide insights into the complex dynamics of gender stereotypes and sexist beliefs within the educational context of Italian high schools, highlighting their multifaceted manifestations and impacts on students and teachers. The coexistence of overt, covert, and subtle forms of sexism ([74]) reveals a deeply entrenched culture that perpetuates traditional gender roles and undermines gender equality.

On the one hand, this study reveals instances of explicit sexist attitudes and behaviors, such as the trivialization of women’s intellectual abilities or the perpetuation of traditional gender roles. These overt manifestations of sexism can have a detrimental impact on the confidence, self-esteem, and educational outcomes of female students (e.g., [23]; [75]). On the other hand, this research also sheds light on more covert and insidious forms of sexism, such as the justification of benevolent sexism or the normalization of sexist microaggressions. These subtle expressions of gender bias can be equally harmful, as they contribute to the perpetuation of a culture that undermines women’s autonomy and agency.

Notably, what emerges from our study is that sexism not only affects women but also has a significant impact on men, creating a range of problems that are often overlooked.

Young men perceive that adhering to sexist attitudes and behavior can have different effects on their lives, both individually and socially. As shown by previous studies ([81]; [68]), sexism affects men’s mental health and relationships and discourages men from expressing emotions, leading them to feelings of isolation and difficulty in connecting with others.

A key finding of this study is the central role teachers and parents play in reinforcing or challenging gender stereotypes.

For instance, gender categorization processes carried out by parents can influence the development of stereotypes that frame certain activities (e.g., play activities) in binary terms. This can exacerbate gender inequality, as children are prevented from engaging in certain activities simply because they are not believed to have the skills associated with their biological sex ([73]). Stereotypes condition women throughout their careers, preventing them from being fairly represented in leadership positions ([6]). Biases affect women in hiring processes, negotiations, job assignments, and performance evaluations. Additionally, they influence women’s behavior, sometimes leading them to self-exclude from leadership roles, experience low self-esteem, or change their demeanor to avoid disapproval ([41]).

To ensure a fair work environment that provides women with equal opportunities for growth and salary, it is essential to recognize how gender norms and societal roles impact women’s presence in leadership positions. Understanding the effect of gender stereotypes would enable the development of interventions at the personal, managerial, and organizational levels to address pay and leadership gaps. Tackling these inequalities would benefit women, their families, male colleagues, organizations, and the global economy.

At the same time, teachers, both male and female, were often identified as agents of sexism, either through their own biased beliefs and behaviors or their failure to address the sexist attitudes and practices prevalent within the school environment.

In particular, the contradiction between the openness of younger generations and the persistence of sexism is noteworthy. The younger generation tends to be more open and gender-aware than previous generations. This was evident in a number of aspects, such as greater sensitivity to gender issues, a greater propensity to recognize and criticize gender-based injustice, and increased acceptance of non-traditional gender roles, with young people supporting gender equality in household and family responsibilities as well as at work.

However, despite these advances, sexism remains deeply rooted and manifests itself in new and subtle ways that drag traditional forms and adapt to social changes. Examples include forms of sexist language ([59]), such as expressions and jokes that perpetuate gender stereotypes; small acts of everyday discrimination that, although they may seem insignificant, accumulate significant negative impact ([16]); and finally, online sexism with digital platforms that are often the theatre of sexist behavior, with comments and attacks targeting women ([14]).

The tension between the openness of new generations and the persistence of sexism reflects a gradual and uneven change. As attitudes are improving, sexism is manifesting itself in new and subtler ways. It is therefore essential to continue working in the field of gender issues, also addressing the issue of new forms of sexism.

Moreover, the shared recognition of these patterns among students and teachers suggests that these perceptions transcend individual experiences, pointing to a broader cultural dynamic. It also highlights the importance of intergenerational dialogue and education in challenging and reshaping these inherited biases, enabling a more inclusive understanding of gender equality across different age groups and socio-cultural settings. This dynamic offers a crucial opportunity for learning and progress, where awareness can spark critical reflection and, ultimately, transformation in societal attitudes.

Furthermore, this study’s exploration of the consequences of sexism for both female and male students and teachers offers a more nuanced understanding of the complex and multifaceted nature of gender inequalities within the educational system.

Notably, this study suggests that the impact of teacher’s gender stereotypes may not be limited to female students but can also shape the experiences and perceptions of their male counterparts.

The data suggest that rigid expectations of masculinity can lead to the marginalization of male students who do not conform to traditional gender norms, potentially hampering their academic and social development. Male students may experience social pressure to conform to rigid models of masculinity and face difficulties in expressing their emotions and vulnerability. This has long-term effects, as even the teachers in our sample, despite being aware of the challenges their male students face in opening up, struggle themselves to support their students in emotional matters.

This is because there are socially desirable expectations associated with being a man, and the violation of these norms may negatively affect mental health since not conforming to traditional norms of masculinity is socially sanctioned ([63]).

Hegemonic masculinity has negative effects on the psychological, social, and physical well-being of men themselves and contributes to perpetuating gender inequalities. Recognizing its limitations and promoting alternative models is essential to building fairer and healthier societies.

This underscores the importance of addressing sexism from a holistic perspective, recognizing its detrimental effects on both genders and seeking to dismantle the harmful assumptions that underlie these beliefs. Once again, implementing comprehensive sex-affective education in schools from a young age would be necessary to deconstruct the stereotypes that oppress both girls and boys.

These stereotypes act as filters through which discriminatory and sexist behaviors are often normalized, perceived as acceptable, or even rendered invisible. In this context, seemingly innocuous but inherently sexist behaviors are frequently underestimated. This minimization stems partly from the difficulty of associating such behaviors with broader power dynamics that underpin discrimination and gender-based violence.

The literature emphasizes how power imbalances in relationships—often justified or unacknowledged—can serve as precursors to abusive behaviors ([3]; [77]). Moreover, the normalization of power imbalances and the entrenchment of gender stereotypes affect how experiences are narrated and perceived.

For example, students and teachers may fail to recognize discriminatory situations they have experienced or observed as problematic because they lack the analytical tools to understand and contextualize them. This limitation is particularly evident in contexts where education on gender issues and violence prevention is underdeveloped. The absence of targeted training programs contributes to limited critical awareness, reducing the ability to distinguish between acceptable behaviors and discriminatory attitudes.

In conclusion, the results of our study, consistent with previous research ([2]; [24]; [71]), can be interpreted from an ecological perspective. This approach allows for analyzing the phenomenon as a complex system that develops across four interconnected levels: individual, relational, community, and social/cultural. This systemic approach enables understanding the dynamics that perpetuate sexism and identifying targeted intervention strategies.

At the individual level, the focus is on personal characteristics and psychological processes that contribute to sexism, such as personal attitudes, internalized gender stereotypes, and implicit or explicit gender biases.

At the relational level, the analysis examines how interpersonal relationships influence the perpetuation of sexism: family dynamics (e.g., division of gender roles), peer relationships that reinforce sexist norms, and power dynamics in intimate or professional settings.

At the community level, the role of communities and local institutions in promoting or countering sexism is analyzed; examples include corporate or school cultures that tolerate sexist attitudes, the lack of female representation in leadership positions, and support structures that inadequately address discrimination (e.g., institutional silence on harassment).

Finally, at the social and cultural level, the focus is on the norms, values, and ideologies prevailing in a society that foster sexism, such as prescriptive gender roles rooted in culture, as well as media and cultural representations that reinforce sexist stereotypes.

## 5. Conclusions

This study provides a nuanced understanding of gender stereotypes and sexist attitudes within Italian high schools, highlighting their pervasive and multidimensional nature. The findings emphasize the pivotal role of teachers in either perpetuating or challenging these biases and underscore the need for targeted interventions to promote gender-sensitive pedagogical practices and critical self-reflection among educators.

This study’s grounded theory approach facilitated a comprehensive understanding of the intricate interplay between gender stereotypes, sexist attitudes, and their consequences in the high school context. Aligned with the ecological model, this study provided a valuable theoretical framework highlighting the multifaceted nature of these phenomena. This model underscored the need for interventions at multiple levels to address the root causes of gender-based inequities effectively. It revealed how deeply embedded stereotypes and biases shape behaviors, attitudes, and structural inequalities, calling for coordinated efforts to foster critical awareness, challenge systemic discrimination, and promote equity at every level of influence.

Adopting an ecological model ([61]) to analyze sexism allows for a deeper understanding of the phenomenon’s complexity and enables integrated interventions. This approach highlights interconnection, as the levels do not operate in isolation but are interdependent. For instance, a sexist culture (social level) can influence corporate norms (community level) and interpersonal relationships (relational level). Furthermore, interventions can be designed at one or more levels to counteract sexism systematically.

The limitations of this study include the specific focus on the Italian context—specifically of a Southern Italian area. As this was a qualitative investigation, the generalizability of the findings was not our primary scope. However, future research might explore the manifestation of these phenomena in other cultural and educational settings, as well as investigate potential intersections with other axes of identity, such as race, class, and sexual orientation.

A further limitation was the gender skew within the study sample, which was disproportionately composed of female participants. This study’s teacher sample was representative of the gender composition of the teaching profession in Italy. However, the inclusion of a greater number of male students would have provided more comprehensive insights into the topic. Future studies should aim to involve a greater number of male participants in this type of research, in order to gain a more holistic understanding of how these issues are experienced by all members of the school community.

Nevertheless, we believe that the findings of this study may have important implications for educational policy and practice.

Specifically, this study’s results highlight the crucial role that teacher attitudes, peer influence, and broader cultural norms play in shaping and maintaining gender stereotypes in educational settings. Considering these findings, it is essential to adopt concrete strategies to counter sexism in schools and promote a more equitable and inclusive environment.

Firstly, this study emphasizes the importance of investing in teacher training and professional development programs that equip educators with knowledge, skills, and resources to create inclusive, gender-sensitive learning environments and promote critical self-reflection on the part of educators.

Indeed, a key step is the introduction of mandatory training programs for teachers and school staff, aimed at raising awareness of gender issues and providing practical tools to counter stereotypes and discrimination. Too often, teachers are unaware of how their expectations, language, and teaching practices can impact students. One of the most effective strategies to tackle gender bias is the integration of gender-sensitive curricula. Teaching materials must be carefully reviewed and, where necessary, revised to eliminate stereotyped representations and promote greater diversity in reference models. In this sense, the concept of gender equality must be addressed in specific disciplines and become a cross-cutting element throughout the training process.

Secondly, our results highlight the need for comprehensive, school-wide interventions that address gender stereotypes and sexist attitudes at multiple levels, targeting not only the students but also the teachers and the broader institutional culture. Creating an inclusive school environment based on mutual respect and non-discrimination can help reduce the social pressure that often forces students to conform to traditional gender roles. To this end, schools should adopt clear regulations against sexism and gender discrimination and promote activities and projects that encourage the overcoming of stereotypes. This can be achieved by fostering non-stereotypical role models and extracurricular activities that challenge gender conventions but also involve families in a shared awareness-raising journey.

In a country where sex-affective education in schools is openly opposed, it is crucial that at least teacher training and university courses address these issues more comprehensively. The focus should be on raising awareness and providing practical strategies for inclusive and empowering teaching practices.

In conclusion, the results of this study suggest that combating gender stereotypes and sexism in educational settings requires a multi-level approach involving students, teachers, educational institutions, and parents. Only coordinated action can foster the creation of educational environments that promote gender equality and well-being for all students. It is essential to adopt a systemic and integrated approach, capable of ensuring a more equitable, inclusive, and respectful school environment for all.

## Figures and Tables

**Table 1 behavsci-15-00230-t001:** Coding process.

Codes	Categories	Macro Categories
Awareness of the existence of inequality between men and women; Smaller gap between men and women today; Tendency to underestimate sexist behavior; Low awareness of the women’s rights movements; Backward socio-cultural context; Confusion/overlap between gender-related issues and issues related to gender identity; Differences in male behavior depending on the socio-cultural context; Generational differences: older adults are more sexist; Having experienced catcalling; Having witnessed gender-based violence; No perceived experience of discrimination.	Evolution of perceptions and experiences regarding gender inequality	Past, present, and future of sexism
Group strategies do not work because men always have to prove something; Total change seen as impossible; School education that leads to the free expression of oneself; School education that leads to the development of critical thinking; Disagreement over changes to the Italian language for gender inclusion; Resolution strategy: individual paths based on communication and information; Resolution strategy: not letting the judgment of others influence oneself; Resolution strategy: increase equality; Resolution strategy: intervene in schools, starting from early childhood; Resolution strategy: awareness campaigns based on empathy (’putting yourself in someone else’s shoes’); Resolution strategies: for older adults (parents-grandparents), there is no hope; Resolution strategy: young people must carry forward new values; Resolution strategy: discussion with friends and family; Resolution strategy: increase equity; Resolution strategy: intervene in schools, starting from early childhood; Resolution strategy: values need to be replaced, as our generation no longer identifies with them; Resolution strategy: raise awareness through demonstrations; Resolution strategy: awareness campaigns based on empathy (’putting yourself in someone else’s shoes’); Resolution strategy: new family education on gender roles.	Proposed solutions to solve gender inequality and sexism	
Gender-based differences in the treatment of children; ‘Short hair is not for girls’; Sexism in the family experienced as a norm, Sexist roles carried on due to cultural tradition, ‘You’re a girl, you should enhance yourself’; Equal responsibility in child-rearing, ‘You’re a woman, so you have to clear the table’; Boys have more freedom to express themselves, ‘You’re a girl, why do you need to drive?’; Men ‘help’ with household chores, Not being able to stay out late or go home alone (without a man), Family’s closed-mindedness about sexuality, Unhealthy communication between father and daughter, No family influence on the mentality of the children, Family influence on the mentality of the children; Household chores are reserved for women.	Gender issue in family	Sexism in microsystems
Teacher’s attitude is different for boys and girls; Teacher more aggressive with girls; Teacher kinder to girls; No gender-differentiated treatment by teachers; Simplified physical tests for girls; Teacher’s perception of boys’ likely interest in gender issues; Difficulty in addressing sexuality in school; Teacher training on sexuality topics; Male teachers’ preference based on the appearance of girls, Lack of opportunities to discuss the topic at school; Teachers’ indifference to the topic, Influence of school on students’ mentality.	Gender issues at school	
Women’s caregiving role in the workplace; Most jobs for women are part-time; Dividing household tasks based on the importance of the job (not gender); Women discriminated against in predominantly male fields, Women discriminated against in the workplace; ‘Women are more attentive at work’.	Gender issues at work	
Men feel obligated to pay the bill, The idea of women as needing protection; Women expect men to pay the bill (at least the first time); Protecting can be seen as chivalry; Men are physically stronger than women; Men and women are different but complementary; Boys avoid discussing certain topics in front of girls.	Benevolent sexism	Forms of sexism
Some sports (soccer, basketball) are more suitable for boys; Very explicit social media content regarding the female body; Influencers serving as female role models to compare with; On social media, women’s bodies are used as objects; Stereotype of boys on social media with defined physiques; Women are better at enduring pain than men; A woman’s attitude can affect the likelihood of being harassed; Women don’t know how to drive; The image of women as lazy; Disparities in the Italian language: some feminine words become insults; Women either work or have children.	Hostile sexism	
Women who post photos in which they are less dressed are judged (especially by other women); Women can be sexist too; Women exaggerate their work-related problem; Women judge the sexual behavior of other women; I won’t have children because I want to become a doctor; Men should help women; “If I had had a daughter, I would never have advised her to follow my educational and career path”; Women are more self-centered than men.	Internalized sexism	
Difficulty going out with a male friend without others judging; Sexist behaviors ‘hurt’ girls; Walking alone in the evening is dangerous; Feeling safe in the street at night when in a group, not necessarily with a man; Women often feel judged for how they dress or do their makeup; Indifference as a defensive reaction to street approaches; Being approached by boys on the street can sometimes be scary; Experiencing sexism lowers self-esteem; Being treated as weak can make one feel weak; Conforming due to judgment; Being discriminated against makes one more indifferent, Internalization of stereotypes as a consequence of discrimination; Dropping out as a consequence of discrimination.	Consequences of sexism for women	Consequences of sexism
Men forced to fit into traditional stereotypes; Men cannot cry, Men must demonstrate their strength and worth; Some males suffer because they are not allowed to show vulnerability; A man who cannot start a fire is looked down upon.	Consequences of sexism for men	
Chivalry is a mask for sexism; ‘I don’t want to be a kept woman’, Family upbringing towards independence, Influencers as symbols of female entrepreneurship; Desire for independence; Equity can increase sexism; Some women can and want to succeed on their own; Feminist struggles should be aimed at not being inferior to men, but not to be superior to them; Work makes women independent; The role of motherhood should not be privileged.	Female empowerment	

## Data Availability

Dataset available on request from the authors.

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
