# Peer review of "Analysis of the Development of Gender Stereotypes and Sexist Attitudes Within a Group of Italian High School Students and Teachers: A Grounded Theory Investigation"

_behavsci, 2025, doi:10.3390/bs15020230_

Round 1
Reviewer 1 Report
Comments and Suggestions for Authors
This is a well-researched and methodologically sound study with strong theoretical grounding and rich qualitative insights. It addresses a critical issue in education—gender stereotypes and sexism—offering valuable perspectives for researchers and practitioners. The use of Grounded Theory is appropriate for exploring complex social phenomena, and the coding structure is well-documented. The paper successfully identifies multiple dimensions of sexism, including hostile, benevolent, and internalized, and presents a nuanced discussion of their effects. It follows a logical structure, moving from a solid literature review to methodology, results, and discussion.
However, the paper would benefit from more concise phrasing and a clearer, more direct writing style. Some sections are overly wordy, making comprehension harder than necessary. Shorter, punchier sentences and stronger topic sentences would improve readability. Additionally, some paragraphs rely too heavily on citation stacking without enough synthesis of the sources. Instead of listing multiple studies, the discussion should briefly summarize their main findings and highlight how they connect to the study’s results.
There are minor inconsistencies in argumentation, particularly regarding generational differences in sexism. The paper acknowledges that younger generations are more open-minded but also claims that sexism remains deeply ingrained. This contradiction should be better framed—are attitudes improving, or are these biases simply manifesting in new ways? A more precise discussion of this tension would strengthen the argument. Furthermore, the paper touches on how sexism affects men but does not fully explore their experiences. Some male students mentioned feeling overlooked in hiring or facing rigid masculinity norms, yet these points are not given much weight in the discussion. A more balanced approach that considers how sexism constrains both genders would add depth to the analysis.
While the study highlights the need for interventions, it lacks concrete recommendations for schools or policymakers. The discussion mentions the importance of teacher training but does not offer specific strategies for reducing gender biases in educational settings. Adding a section outlining practical steps—such as integrating gender-sensitive curricula, implementing training programs for educators, or addressing sexist behaviors in schools—would make the paper more actionable.
There is some repetition in the results section, particularly when discussing sexism in microsystems. Certain points, especially regarding teacher behavior and family influence, are restated in different parts of the paper. Streamlining these sections would improve flow without losing depth. Minor stylistic and formatting issues also need attention, such as occasional awkward phrasing, inconsistencies in terminology (switching between “gender stereotypes” and “sexist beliefs” without clear distinction), and long paragraphs that could be broken up for readability.
Overall, this is a strong study with a well-structured analysis and valuable contributions to the field.
Comments on the Quality of English Languagenone
Author Response
Reviewer 1
This is a well-researched and methodologically sound study with strong theoretical grounding and rich qualitative insights. It addresses a critical issue in education—gender stereotypes and sexism—offering valuable perspectives for researchers and practitioners. The use of Grounded Theory is appropriate for exploring complex social phenomena, and the coding structure is well-documented. The paper successfully identifies multiple dimensions of sexism, including hostile, benevolent, and internalized, and presents a nuanced discussion of their effects. It follows a logical structure, moving from a solid literature review to methodology, results, and discussion.
However, the paper would benefit from more concise phrasing and a clearer, more direct writing style. Some sections are overly wordy, making comprehension harder than necessary. Shorter, punchier sentences and stronger topic sentences would improve readability.
We wanted to thank the reviewer for taking the time to read our manuscript. We understand the time commitment involved in reviewing a manuscript, and we sincerely thank you for your careful consideration of our work, and for your valuable feedback. We have now revised our manuscript according to your comments. Highlighted in green you will find the requested new addition to the text; MS Word revision function was used to highlight the new phrasing and language revision. We also replied point-by-point to your comments here:
Additionally, some paragraphs rely too heavily on citation stacking without enough synthesis of the sources.
We now did it
Instead of listing multiple studies, the discussion should briefly summarize their main findings and highlight how they connect to the study’s results.
We now did it
- There are minor inconsistencies in argumentation, particularly regarding generational differences in sexism. The paper acknowledges that younger generations are more open-minded but also claims that sexism remains deeply ingrained. This contradiction should be better framed—are attitudes improving, or are these biases simply manifesting in new ways? A more precise discussion of this tension would strengthen the argument.
We have now broadened the discussion regarding generational differences in sexism.
- Furthermore, the paper touches on how sexism affects men but does not fully explore their experiences. Some male students mentioned feeling overlooked in hiring or facing rigid masculinity norms, yet these points are not given much weight in the discussion. A more balanced approach that considers how sexism constrains both genders would add depth to the analysis.
We have now broadened the discussion about the consequences of sexism on men.
- While the study highlights the need for interventions, it lacks concrete recommendations for schools or policymakers. The discussion mentions the importance of teacher training but does not offer specific strategies for reducing gender biases in educational settings. Adding a section outlining practical steps—such as integrating gender-sensitive curricula, implementing training programs for educators, or addressing sexist behaviors in schools—would make the paper more actionable.
Thanks for this suggestion. We have now added some practical recommendations.
- There is some repetition in the results section, particularly when discussing sexism in microsystems. Certain points, especially regarding teacher behavior and family influence, are restated in different parts of the paper. Streamlining these sections would improve flow without losing depth.
We tried to eliminate some possible repetitions
Minor stylistic and formatting issues also need attention, such as occasional awkward phrasing, inconsistencies in terminology (switching between “gender stereotypes” and “sexist beliefs” without clear distinction), and long paragraphs that could be broken up for readability.
We now made a throughout revision in order to address your comment
Overall, this is a strong study with a well-structured analysis and valuable contributions to the field.
Thanks again!
Reviewer 2 Report
Comments and Suggestions for Authors
The manuscript “Analysis of the development of gender stereotypes and sexist attitudes within a group of Italian high school students and teachers: A Grounded Theory investigation” focuses on an important and interesting topic. Overall, the manuscript is very well written and clear. Therefore, I suggest that it be accepted for publication. However, I have some suggestions for improvement.
1. Introduction – The Introduction is very well written and clear, there is only one mistake as there are two sections 1.3.: “Sexism in School context” and “The Gender Perspective in Initial Teacher Training”.
2. Materials and Methods – First, I would like to say that instead of the existing 3 subsections in this section, I would prefer to see the traditional 4 subsections divided as follows:
Participants (where the sample must be described);
Procedure (where the entire study procedure must be described; here you should include the sentence about your institution's Ethics Committee);
Instrument (where all data collection material or instruments must be described); and
Data analysis (where it is stated how the data will be analysed). This way this section will become clearer, and it will be possible to replicate the study.
Second, I suggest that in the current subsection “2.1. Participants and procedure”, after the average age of the teachers, the Standard Deviation should also be inserted in parentheses.
The second paragraph of the subsection "2.3. Data analysis" ends by saying "which can summarise the results that emerged from the data.” I would like to emphasize that the results do not emerge from anywhere; researchers have an active role in the analysis, so I suggest you say that the results were identified in the focus group material.
I make the same suggestion for the following paragraph of this same subsection. I wouldn't say "emerging categories," because the categories were identified by the study's researchers. Right?
In the last paragraph of this subsection, it says that “To ensure the reliability of the data analysis, the research team discussed the codes and categories attributed, involving both the main researchers and the PhD students who acted as observers”, but it is not clear whether all these people are authors of this manuscript.
Was the focus group guide/script or grid the same for students and teachers? Please clarify.
Here, it should also be made clear how the material from the focus groups was analysed. For example, was the material from the student focus groups analysed together with the material from the teacher focus groups or was it analysed separately?
I also suggest that you assign a code to each participant, so that each excerpt from the focus group referenced in the Results section is properly identified (e.g., participant 1, from student focus group 1, I suggest something like: (P1, FG1, female student).
3. Results - In the first paragraph of the results, I was very curious to know who contributed more to the discourses identified in the focus group material (i.e., for each category): the students or the teachers? The male or female participants? I think the results would be richer if this information was added.
In the results section you refer to “emerged” several times (e.g., in the first paragraph you say “the information that emerged from the interviews”) and I suggest you refer instead, for example, “the information was identified in the material from the interviews" (or focus groups).
Regarding Table 1, I think that the designation/title of some categories could be improved in order to be a better umbrella for the respective codes. For example, the way the “Proposed solutions” category has been named doesn't tell the reader much, it would be better if you could make the leap and give us a better idea of what these “Proposed solutions” are or what areas or contexts the “Proposed solutions” refer to. Why not something like: “Proposed solutions to solve sexism”?
Throughout the description of the categories, you will highlight some excerpts from the participants in the focus groups that are not identified. I suggest that you refer to the code you have defined for each participant (e.g., referring to the focus group number, whether they are a student or a teacher, whether they are male or female, etc.).
Note that sometimes you put the participants' excerpts in italics and sometimes you don't. You must homogenize the excerpts.
Personally, I never present the results like you did. I usually present the themes/categories first and only then insert excerpts from the focus groups to illustrate them. But I believe there are several possibilities for us to present results.
4. Discussion - The discussion and conclusion are quite good and respond to the objectives of the study.
Author Response
Reviewer 2
The manuscript “Analysis of the development of gender stereotypes and sexist attitudes within a group of Italian high school students and teachers: A Grounded Theory investigation” focuses on an important and interesting topic. Overall, the manuscript is very well written and clear. Therefore, I suggest that it be accepted for publication. However, I have some suggestions for improvement.
We wanted to thank the reviewer for taking the time to read our manuscript. Thank you for your thoughtful and constructive feedback. Your comments have been very helpful in improving our manuscript. We have now revised our manuscript according to your comments. Highlighted in green you will find the requested new addition to the text; MS Word revision function was used to highlight the new phrasing and language revision. We also replied point-by-point to your comments here:
- Introduction – The Introduction is very well written and clear, there is only one mistake as there are two sections3.: “Sexism in School context” and “The Gender Perspective in Initial Teacher Training”.
We have now corrected the error.
- Materials and Methods – First, I would like to say that instead of the existing 3 subsections in this section, I would prefer to see the traditional 4 subsections divided as follows:
Participants (where the sample must be described);
Procedure (where the entire study procedure must be described; here you should include the sentence about your institution's Ethics Committee);
Instrument (where all data collection material or instruments must be described); and
Data analysis (where it is stated how the data will be analysed). This way this section will become clearer, and it will be possible to replicate the study.
We have divided the sections as required
- Second, I suggest that in the current subsection “1. Participants and procedure”, after the average age of the teachers, the Standard Deviation should also be inserted in parentheses.
We added Standard Deviation
- The second paragraph of the subsection "3. Data analysis" ends by saying "which can summarise the results that emerged from the data.” I would like to emphasize that the results do not emerge from anywhere; researchers have an active role in the analysis, so I suggest you say that the results were identified in the focus group material.I make the same suggestion for the following paragraph of this same subsection. I wouldn't say "emerging categories," because the categories were identified by the study's researchers. Right?
Thank you for this comment. Indeed, using the term "identify" is appropriate, as it acknowledges that researchers play an active role in making sense of the data. Therefore, it is more relevant to say that the categories are "identified" or "constructed" by the researchers based on the analysis of the empirical material.
In the last paragraph of this subsection, it says that “To ensure the reliability of the data analysis, the research team discussed the codes and categories attributed, involving both the main researchers and the PhD students who acted as observers”, but it is not clear whether all these people are authors of this manuscript.
The team that discussed the codes and categories consists of the authors and other researchers (who we have mentioned in the acknowledgements section). We have now made that clear in the manuscript.
- Was the focus group guide/script or grid the same for students and teachers? Please clarify.
We clarified this information
- Here, it should also be made clear how the material from the focus groups was analysed. For example, was the material from the student focus groups analysed together with the material from the teacher focus groups or was it analysed separately?
We added this information
I also suggest that you assign a code to each participant, so that each excerpt from the focus group referenced in the Results section is properly identified (e.g., participant 1, from student focus group 1, I suggest something like: (P1, FG1, female student).
We now did it.
- Results –
In the first paragraph of the results, I was very curious to know who contributed more to the discourses identified in the focus group material (i.e., for each category): the students or the teachers? The male or female participants? I think the results would be richer if this information was added.
We now did it.
- In the results section you refer to “emerged” several times (e.g., in the first paragraph you say “the information that emerged from the interviews”) and I suggest you refer instead, for example, “the information was identified in the material from the interviews" (or focus groups).
We replaced the verb “emerge” with more suitable ones.
- Regarding Table 1, I think that the designation/title of some categories could be improved in order to be a better umbrella for the respective codes. For example, the way the “Proposed solutions” category has been named doesn't tell the reader much, it would be better if you could make the leap and give us a better idea of what these “Proposed solutions” are or what areas or contexts the “Proposed solutions” refer to. Why not something like: “Proposed solutions to solve sexism”?
We renamed the category according to your suggestion.
Throughout the description of the categories, you will highlight some excerpts from the participants in the focus groups that are not identified. I suggest that you refer to the code you have defined for each participant (e.g., referring to the focus group number, whether they are a student or a teacher, whether they are male or female, etc.).
We now did it.
- Note that sometimes you put the participants' excerpts in italics and sometimes you don't. You must homogenize the excerpts.
We now homogenised all excerpts
Personally, I never present the results like you did. I usually present the themes/categories first and only then insert excerpts from the focus groups to illustrate them. But I believe there are several possibilities for us to present results.
- Discussion - The discussion and conclusion are quite good and respond to the objectives of the study.
Thanks again!